# The Influence of Confined Space Size on the Temperature Distribution Characteristics of Internal Window Plume from Well-Ventilated Compartment Fires

**Qiwei Dong** [1,2], **Yanfeng Li** [1,*], **Junmei Li** [1,*], **Fei Xie** [2], **Desheng Xu** [1] **and Zhihe Su** [1]

[1] Beijing Key Laboratory of Green Built Environment and Energy Efficient Technology, Beijing University of Technology, Beijing 100124, China; dongqiwei@emails.bjut.edu.cn (Q.D.); xusimple@emails.bjut.edu.cn (D.X.); suzh@emails.bjut.edu.cn (Z.S.)

[2] Beijing General Municipal Engineering Design & Research Institute, Beijing 100082, China; codyfei@bmedi.cn

[*] Correspondence: liyanfeng@bjut.edu.cn (Y.L.); lijunmei@bjut.edu.cn (J.L.); Tel.: +010-67391608 (Y.L. & J.L.)

**Abstract:** In this research, the influence of confined space size on the temperature distribution characteristics of internal window plumes from well-ventilated compartment fires was studied. Theoretical analysis was firstly used to establish a mathematical model for the smoke after ejecting from the window in the space. The study considered fire heat release rate and vertical height as dependent variables. Numerical simulations and experimental methods were carried out to study the temperature variations. A critical distance $L_2$ was obtained. Results show that when the space $D$ between the vertical retaining wall and the building façade is greater than $L_2$, the variation of $D$ has little influence on radial temperature. Once $D$ is less than $L_2$, the radial temperature distribution inside the confined space will tend to be consistent, and the temperature in the confined space sharply increases as $D$ decreases. In addition, a dimensionless model was derived to quantify the relationship between temperature rise and vertical height. The experimental and numerical simulation results were processed, which are in good agreement with the model. The study can provide a framework for managing building safety.

**Keywords:** confined space; temperature distribution; window plume; well-ventilated compartment fires

## 1. Introduction

Nowadays, high-rise building fires often occur, causing serious casualties [1–4]. The fire can spread vertically through the inner vertical shaft, atrium, or through the external windows and openings on the building's exterior facade [5,6]. In modern high-rise buildings, with more traditional concrete exterior walls being replaced by glass curtain walls, the gaps between glass curtain walls and floor slabs or the channels of double-shin facades have added new ways for the vertical spread of fire [7–9]. Currently, almost every city has a large number of commercial streets, which, due to the narrow width of the streets, can form street valleys. When a fire occurs on one side of the building, it may have an impact on the buildings on the other side of the street. The above two types of buildings have a common feature, belonging to channel type of confined spaces. In a fire scenario, smoke overflows through the window, and its subsequent flow might be restricted by solid boundaries, such as vertical and retaining walls on both sides. The temperature distribution characteristics and smoke movement in such buildings are limited by the space width and length. In addition, the spread pattern and impact range of plumes are different from those in open spaces. Therefore, it is necessary to study these important issues.

Due to the complexity of the fire, there are many factors that influence window plumes in confined spaces. Oleszkiewicz et al. [10–13] conducted a full-scale experiment to study the influence of vertical retaining wall on the window fire plume. Qualitative research was conducted on the changes in plume morphology, flame, and radiation intensity. The

research results indicate that window plumes are more likely to overflow from wider windows. When the window width is small, the heat flow formed by the exterior facade is small. Delichatsios and Lee Yee-ping [14,15] introduced a feature length and studied the effect of vertical barriers on the ejecting flame height. They proposed a flame height correction model when the distance between vertical barriers is less than the feature's length. Tang et al. [16,17] investigated the building facade ejecting fire behavior under different external boundaries. A series of fire experiments were carried out to obtain parameters of flame ejecting behavior, and a dimensionless prediction model was established to describe flame extension. Lu et al. [17] employed experiments to investigate the side wall effects on flame ejecting behavior. Two side walls were positioned symmetrically at the two sides of the compartment window. The compartment ventilation and side wall constraint conditions were varied by changing the window dimensions and the side wall separation distances. Dimensionless characteristic parameter theoretical prediction models were brought up. Akito et al. [18,19] conducted small-scale experiments to investigate the effect of vertical retaining walls on the temperature distribution outside the window. They found that the plume shape outside of the window would change when the distance from the vertical retaining was small.

Generally, building room firess can be divided into under-ventilated and well-ventilated compartment fires based on different ventilation conditions and fire loads [20]. The heat release rate of the fire (the fire HRR) in a well-ventilated compartment is related to the combustion degree and fire load density. The fire room is considered to be in two regions, namely, the dual region model. The upper layer in the room is the hot smoke layer, and the lower layer is the cold air layer. It is believed that the boundary height between the two layers is consistent at all positions, and the parameters such as temperature, pressure, density, and smoke concentration inside the two layers are also uniform [20]. The heat release rate of the under-ventilated compartment fires is mainly directly related to the ventilation factor and combustion degree. Under-ventilated compartment fires are generally in the newly ignited room where flashover occurs [20]. In this condition, internal combustion is intense, and it can be taken as a single area with only a high-temperature smoke layer. The type of fire combustion is usually determined by fire HRR. For under-ventilated conditions, $Q$ is more than $1500A\sqrt{H}$. For well-ventilated conditions, $Q$ is less than $1500A\sqrt{H}$ ($Q$ is the fire HRR in the combustion chamber, $A$ is the opening area, $H$ is the opening height, same as below).

In summary, it is found that most of the previous research is based on under-ventilated compartment fires. For modern large office buildings, there are large ventilation openings. The flow field and heat transfer processes are complex, and the main longitudinal spread along the building is hot smoke with no flame ejection. However, few theoretical analyses can be developed for such fires [21], which has always been one of the difficulties in building fires. Miao et al. [22] conducted experiments and numerical simulations to study window plumes from a room fire to the cavity of a double-skin façade. Some of their conditions are based on well-ventilated compartment fires. The objective of the study is to determine the conditions under which the plume would attach to the interior or to the exterior skin. Firstly, a simple theoretical model was proposed to calculate the net lateral pressure on the plume. Then, experimental and numerical simulation studies were conducted. Both experimental and simulation results showed that for a given cavity width, heat release rate and geometry of the window opening were the two key factors in determining the plume trajectory. Li and Bai [23,24] studied the characteristics of window plumes based on the under-ventilated compartment fires. But, their research focused on the side wind effect on the flow behavior of the window plume under free working conditions. Therefore, further research is needed to investigate the general laws in the propagation characteristics of window plumes in channel-confined spaces under such combustion types.

This research aims to investigate the impact of confined channel space on the temperature distribution of internal window plumes under well-ventilated compartment fires. Theoretical analyses were first used to establish a mathematical model for the smoke after

overflowing from the window in the space. Then, numerical simulation and experimental research were carried out to study the general law of the spread of smoke in confined spaces. A critical distance $L_2$ was obtained, which can provide a basis for the width of confined spaces. In addition, a dimensionless model was developed to quantify the relationship between temperature rise and vertical height, which can measure the degree of harm of combustion chamber fires to superstructure when the width of confined spaces is small. The simulation and experimental results are consistent with the model. These findings mentioned above provide a basis for building fire prevention outside windows.

## 2. Theoretical Analysis

In order to analyze the impact of confined space size on the propagation characteristics of internal window plumes, this study first theoretically analyzed the propagation mechanism of plumes. Figure 1 presents a diagram of window plume propagation. After a fire occurs in the combustion chamber, the smoke ejects through the opening and transitions from horizontal to vertical due to the buoyancy force. The length $L_2$ in which the smoke transitions from horizontal to vertical depends on the competitive balance between the horizontal momentum and vertical buoyancy of the smoke.

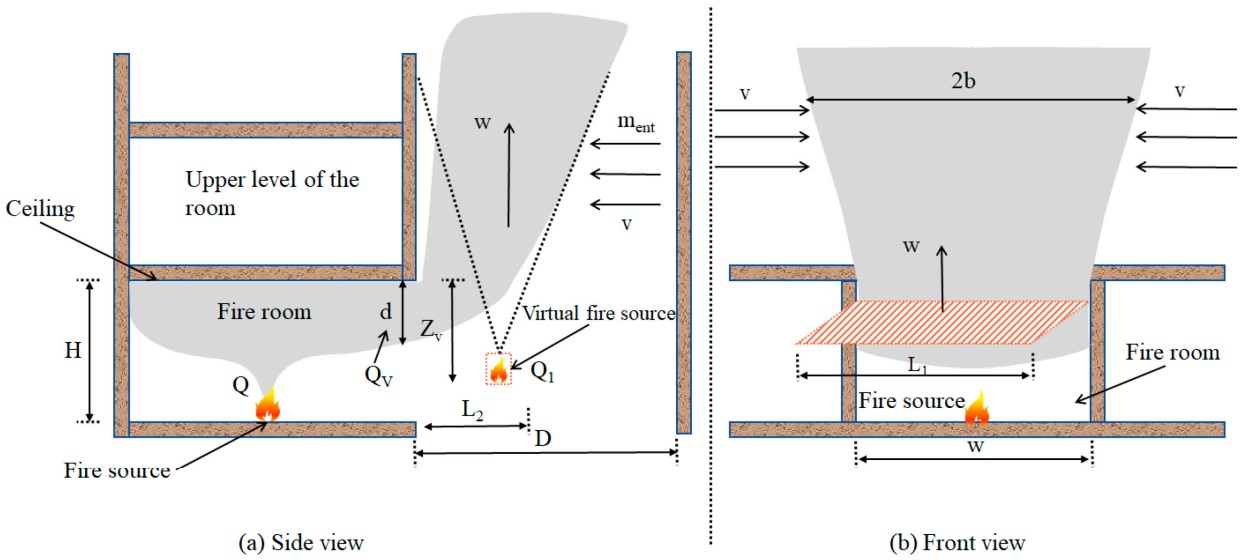

(a) Side view        (b) Front view

**Figure 1.** Schematic diagram of window plume propagation.

When the ejecting smoke changes from horizontal to vertical through the opening, the airflow rate entrained into the plume is [25,26]:

$$m_{ent} = \rho_\infty \left( \sqrt{\frac{\Delta T_f}{T_\infty} g L_2} \right) L_2 L_1 \tag{1}$$

where $\rho_\infty$ is the air density, $\Delta T_f$ is the smoke temperature rise, $T_\infty$ is the ambient temperature, $g$ is the gravitational acceleration, $L_2$ is the distance from the horizontal direction to the vertical direction when the smoke overflows from the opening, $L_1$ is the feature's length [27], which represents the length of the virtual rectangular heat source parallel to the exterior facade of the building in an open flow.

Due to the entrainment of surrounding air, the upward momentum of smoke ejecting through the opening can be expressed as:

$$M_{ent} = m_{ent} w \tag{2}$$

where $w$ is the upward velocity of the ejecting plume. By substituting Equation (1) into Equation (2), the upward momentum can be written as:

$$M_{ent} \propto \rho_\infty \left( \sqrt{\frac{\Delta T_f}{T_\infty} g L_2} \right) L_2 L_1 \left( \sqrt{\frac{\Delta T_f}{T_\infty} g L_2} \right) \tag{3}$$

The horizontal momentum of smoke at the opening can be expressed as:

$$M_0 \approx \rho_g \frac{\Delta T_g}{T_\infty} (H - Z_0)^2 g W \tag{4}$$

where $\rho_g$ is the air density, $W$ is the opening width, $\Delta T_g$ is the temperature rise in the combustion chamber, $Z_0$ is the distance from the neutral surface to the bottom of the opening.

$L_2$ mainly depends on the competitive balance between horizontal momentum and buoyancy after smoke overflow:

$$M_{ent} \propto \rho_\infty \frac{\Delta T_f}{T_\infty} g L_2^2 L_1 \approx \rho_g \frac{\Delta T_g}{T_\infty} (H - Z_0)^2 g W = M_0 \tag{5}$$

By substituting $L_1 = (A\sqrt{H})^{2/5}$ into the above equation, $L_2$ can be expressed as:

$$L_2 \propto \left( \frac{\Delta T_g}{\Delta T_f} \right)^{1/2} \left( \frac{\rho_g}{\rho_\infty} \right)^{1/2} \left( 1 - \frac{Z_0}{H} \right) \left( A H^{4/3} \right)^{3/10} \tag{6}$$

After the plume spreads through the window, the decrease in vertical temperature inside the confined space is due to the radial and lateral entrainment of the surrounding air.

If a vertical retaining wall is placed outside of the building facade, a passage will be formed by the building facade and vertical retaining wall. When the distance between the vertical retaining wall and the building façade $D$ (the distance $D$) is less than $L_2$, the radial entrainment will be completely restricted, leaving only the lateral entrainment on both sides. This study introduces the concept of a virtual ignition source. Based on a previous research [28], $Zv = (d + M_0)/0.15Q^{1/3}$, where $Z_v$ is the distance from the virtual ignition source to the upper edge of the overflow plume, and $d$ is the thickness of the smoke layer in the combustion chamber.

When the distance $D$ is less than $L_2$, assuming that the velocity of air entrained on both sides of the plume $v = \alpha w$, the internal ignition source plume model in a confined space is derived.

Where $\alpha$ is a constant and is taken to be 0.51 [29], $2b$ is the width after the plume ejects from the window.

Consider a micro element in the vertical direction of the internal plume in a confined space, as shown in Figure 2. According to the mass balance of the control volume, the mass flow rate through plane $z$ ($z$ is the height from a certain point to the virtual ignition source, $z = Z - H + Z_V$) is:

$$m_z = \int_{-\infty}^{+\infty} \rho_g w \cdot D dy = \int_{-b}^{b} \rho_g w \cdot D dy = 2 \rho_g w \cdot D b \tag{7}$$

The mass flow rate of air entrainment through both sides is:

$$dm_z = m_b = 2 \rho_\infty v \cdot D dz \tag{8}$$

The mass flow rate through plane $z + dz$ is:

$$m_{z+dz} = m_z + \frac{dm_z}{dz} dz \tag{9}$$

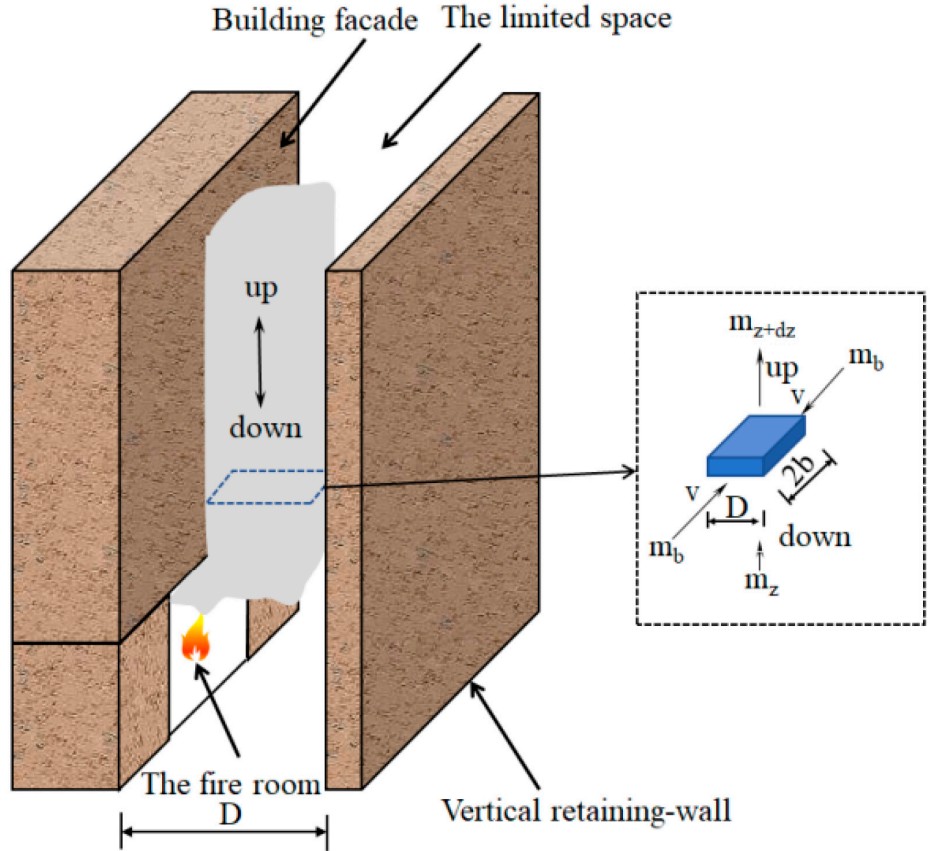

**Figure 2.** Internal micro-element model of a confined space.

According to the Boussinesq approximation, the variation in density is small relative to the ambient reference density everywhere except in the buoyancy term. By substituting Equations (7) and (8) into Equation (9), it can be obtained:

$$\frac{d(wb)}{dz} = \alpha w \tag{10}$$

According to the momentum balance of the control volume, the momentum at plane $z$:

$$\vec{M}_z = \int_{-\infty}^{\infty} \rho_g w^2 \cdot D dy = \int_{-b}^{b} \rho_g w^2 \cdot D dy = 2\rho_g w^2 Db \tag{11}$$

At plane $z + dz$:

$$\vec{M}_{z+dz} = \vec{M}_z + \frac{d\vec{M}_z}{dz} dz \tag{12}$$

The upward force acting on the control volume should be equal to:

$$\vec{F} = 2(\rho_\infty - \rho_g)g \cdot Db \cdot dz \tag{13}$$

From the conservation of momentum of the controlled volume, the upward force can be expressed as:

$$\vec{F} = d\vec{M}_z \tag{14}$$

By substituting Equations (11) and (13) into Equation (14), it can be obtained:

$$\frac{d(w^2 b)}{dz} = \frac{\Delta\rho}{\rho_g} gb \tag{15}$$

where $\Delta\rho = \rho_\infty - \rho_g$, by using the ideal gas law:

$$\frac{\Delta\rho}{\rho_g} = \frac{\Delta T}{T_\infty} \tag{16}$$

where $\Delta T = T - T_\infty$, by substituting Equation (16) into Equation (15):

$$\frac{d(w^2 b)}{dz} = \frac{\Delta T}{T_\infty} gb \tag{17}$$

Energy flux at plane $z$:

$$Q_z = \int_{-\infty}^{\infty} \rho_\infty C_p w \Delta T \cdot D dy = \int_{-b}^{b} \rho_\infty C_p w \Delta T \cdot D dy = 2\rho_\infty C_p w \Delta T \cdot bD \tag{18}$$

where $C_P$ represents the specific heat capacity of the gas. Energy flux through plane $z + dz$:

$$Q_{z+dz} = Q_z + \frac{dQ_z}{dz} dz \tag{19}$$

Considering the outer boundary of the plume as adiabatic, according to the law of energy conservation, it can be obtained that:

$$d(w\Delta T b) = 0 \tag{20}$$

Integrating Equation (20):

$$b\Delta T = \frac{K}{w} \tag{21}$$

By substituting Equation (21) into Equation (17):

$$\frac{d(w^2 b)}{dz} = \frac{gK}{T_\infty} \cdot \frac{1}{w} \tag{22}$$

where $K$ is an unknown constant, since the vertical velocity and temperature difference is only the function of height, therefore it can be assumed that:

$$b = Y_1 \cdot z^{s_1} \tag{23}$$

$$w = Y_2 \cdot z^{s_2} \tag{24}$$

$$\Delta T = Y_3 \cdot z^{s_3} \tag{25}$$

Among them, $Y_1$, $Y_2$, $Y_3$, and $s_1$, $s_2$, $s_3$ are constants. Substitute Equations (23) and (24) into Equation (10):

$$\frac{d}{dz}(Y_2 Y_1 z^{s_2+s_1}) = \alpha Y_2 z^{s_2} \tag{26}$$

By integrating Equation (26) and comparing the index of $z$:

$$s_1 = 1 \tag{27}$$

By substituting Equations (23) and (24) into Equation (22), it can be obtained that:

$$\frac{d}{dz}(Y_2^2 Y_1 z^{2s_2+s_1}) = gk\frac{z^{-s_2}}{Y_2 T_\infty} \tag{28}$$

$$s_2 = 0 \tag{29}$$

$$Y_1 = \alpha/(1+s_2) = \alpha \tag{30}$$

$$Y_2 = [\frac{gK}{T_\infty} \cdot \frac{1}{\alpha}]^{1/3} \tag{31}$$

Therefore, half the width after the plume ejects from the window $b$, and the upward velocity of the ejecting plume w can be expressed as follows:

$$b = \alpha z \tag{32}$$

$$w = [\frac{gK}{T_\infty} \cdot \frac{1}{\alpha}]^{1/3} \tag{33}$$

The fire HRR $Q_1$ of the virtual ignition source can be calculated by:

$$Q_1 = \int_{-b}^{b} \rho_\infty C_p w \Delta T D dy = 2\rho_\infty C_p w \Delta T b D \tag{34}$$

$$K = \frac{Q_1}{2\rho_\infty C_p D} \tag{35}$$

Previous researchers have verified this through experiments. Due to the adiabatic outer boundary of the plume, $Q_1$ is equal to the heat release rate $Q_V$ overflowing from the window [29].

$$Q_1 = \frac{2\sqrt{2g}}{3}(k_Q/k_m)\rho_g C_p d^{1.5} W \Delta T_g^{1.5} T_\infty^{-0.5} \tag{36}$$

where $K_Q$ and $K_m$ is the profile factor coefficients, and are taken to be 0.9 and 1.35 [29]. By substituting Equation (35) into Equation (33):

$$w = [\frac{gQ_1}{2\rho_\infty C_p D T_\infty \alpha}]^{1/3} \tag{37}$$

By substituting Equations (23) and (25) into Equation (21):

$$Y_1 Y_3 z^{s_1+s_3} = K \frac{z^{-s_2}}{Y_2} \tag{38}$$

It can be obtained that:

$$S_3 = -1 \tag{39}$$

$$Y_3 = \frac{K}{Y_1 Y_2} \tag{40}$$

By substituting Equations (30), (31), (35), (39), and (40) into Equation (25), the temperature rise $\Delta T$ can be expressed as:

$$\Delta T = (\frac{Q_1}{2\rho_\infty C_p \alpha D})^{2/3} \cdot (\frac{T_\infty}{g})^{1/3} z^{-1} \tag{41}$$

## 3. Experiments

This chapter mainly introduces the experimental research on the influence of confined space size on the temperature distribution characteristics of internal window plumes from well-ventilated compartment fires. Among them, Section 3.1 introduces the construction of the scaled experimental platform, and Section 3.2 introduces the main conclusions obtained from the experiment.

### 3.1. Experimental Setup

This study built a 1:8 scaled experimental platform based on similarity criterion. The main body of the model experimental platform was 2.8 m long, 0.6 m wide, and 1.2 m high. The main structure was made of galvanized steel plate. The main body of the experimental platform was designed into three layers. The bottom layer is the combustion

chamber, which consists of rooms of two different sizes. The length, width, and height of the small combustion chamber are 0.5 m, 0.5 m, and 0.4 m, respectively. The length, width, and height of the large combustion chamber are 1 m, 0.5 m, and 0.4 m, respectively. The surrounding walls of the combustion chamber and adjacent room walls were covered with fireproof cotton for fire prevention and insulation. The combustion chamber was equipped with adjustable windows, and the window size could be changed by pushing and pulling the fireproof board. In addition, the combustion chamber can be supplemented with air through the additional tuyere at the rear of the experimental platform. The second part of the model experimental platform is a vertical retaining wall, which is mainly composed of fire dampers and supports. The length and height of the vertical retaining wall correspond to the main body of the experimental platform, and a fireproof glass is installed in the central part of the retaining wall as an observation window to observe the combustion inside the combustion chamber. The distance $D$ can be changed by easily moving the vertical retaining wall. As shown in Figure 3.

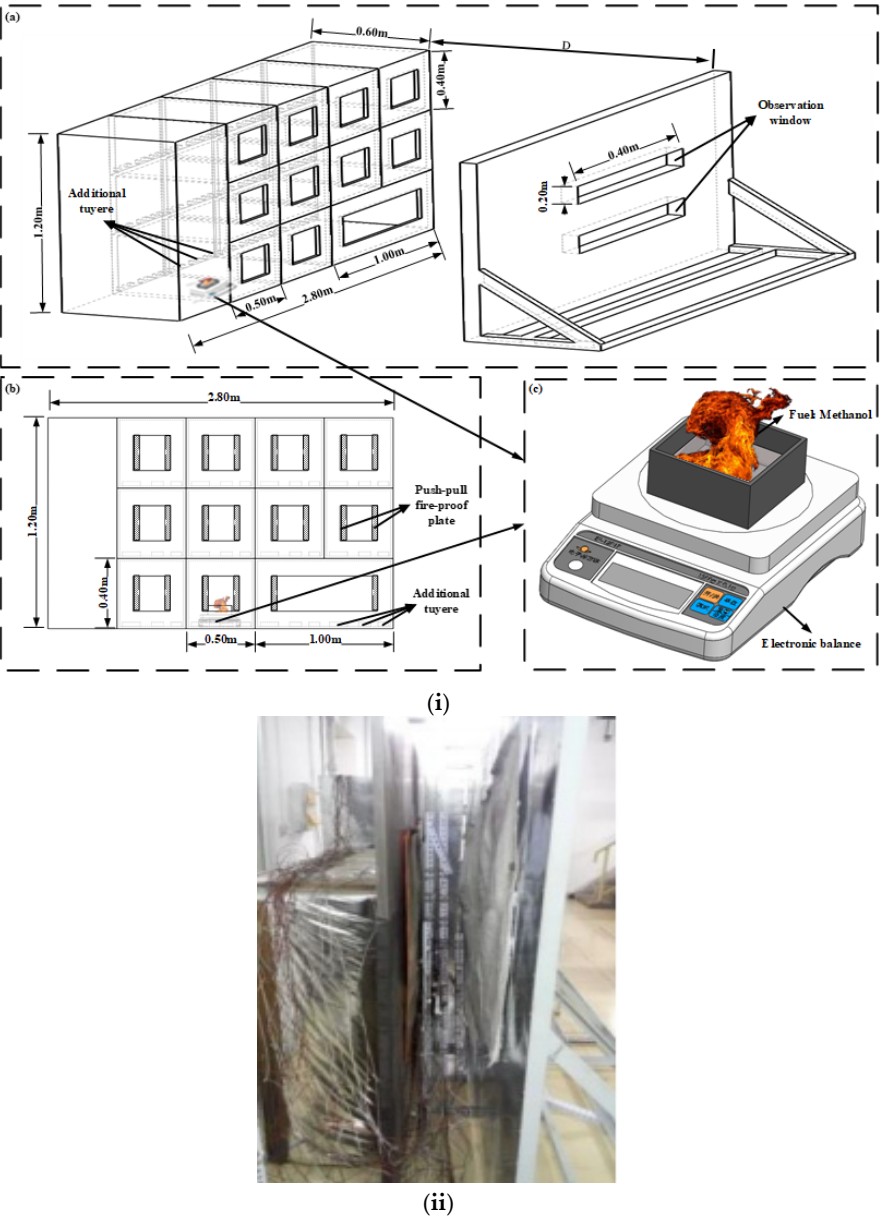

(i)

(ii)

**Figure 3.** Small-scale model of narrow and confined space window plume experiments, (**i**) diagram of the small-scale model, (**a**) Overall schematic diagram, (**b**)Front view of building façade, (**c**) Electronic balance and fuel, and (**ii**) actual photo of the experiments.

Figure 4 shows the side view and top view of the measurement point arrangement of the temperature measurement system inside the confined space. A total of six sets of thermocouples were arranged inside the confined space, with two sets closest to the combustion chamber, each set containing nine thermocouples, two sets of thermocouples in the middle, each set containing six thermocouples, the outermost two sets of thermocouples, each set containing two thermocouples. The horizontal spacing of each thermocouple is 8 cm, and the vertical spacing is 12 cm. The number of thermocouple sets along the radial direction of the building facade also decreases as the distance $D$ decreases.

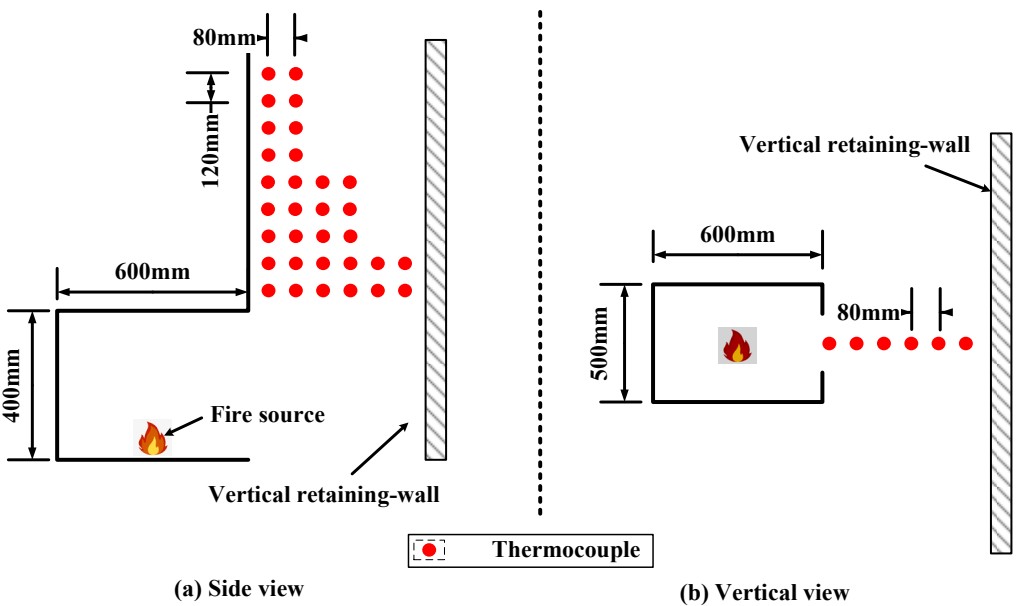

**Figure 4.** The layout of measurement points for temperature measurement system.

The combustion pool in the study used fuel discs, each with a combustion area of 0.01 m². By combining different numbers of fuel discs, the change in fuel combustion area can be achieved. The mass measurement system used in the experiment was an electronic balance, and the combustion rate of the combustion pool was calculated by recording the mass changes of the electronic balance. The temperature measurement system used in the experiment was K-type thermocouples and T-type thermocouples with 0.5 mm diameter, which were used to measure the temperature changes inside the confined space.

There were two types of fuel pool combustion areas designed in the experiment: one is 0.1 m × 0.1 m, and the other is 0.1 m × 0.2 m. There were two fuels used in the experiment, one is methanol, and the other is n-heptane. The fire HRR $Q$ was calculated using the mass loss method [30]. $Q = \eta \times m \times h_c$, $\eta$ is the combustion efficiency, reflecting the degree of incomplete combustion of combustibles, with a value of 0.3–0.9. $m$ is the mass combustion rate of combustible materials. $h_c$ is the average calorific value of the combustible material. The calorific value of methanol is 22,703 kJ/kg, and the calorific value of n-heptane is 44,600 kJ/kg. After calculation, the fire HRR $Q$ of the three working conditions are 6.81, 9.08, and 17.84 kW, respectively. As shown in Table 1.

**Table 1.** Test conditions and calculation results of the fuel thermal properties of the oil pool.

| Working Condition | Fuel | Fuel Pool Size (m²) | Mass Combustion Rate (kg/s·m²) | Fire HRR $Q$ (kW) |
|---|---|---|---|---|
| SF1 | methanol | 0.1 × 0.1 | 0.03 | 6.81 |
| SF2 | methanol | 0.1 × 0.2 | 0.02 | 9.08 |
| SF3 | n-heptane | 0.1 × 0.1 | 0.04 | 17.84 |

The width of the building facade opening $W$ in the experiment were 0.2 m, 0.3 m, and 0.5 m, respectively. The height $H$ was 0.4 m. By changing the distance $D$, the size of the confined space was adjusted. Four types of distance $D$ were set, in descending order of 32 cm, 24 cm, 16 cm, and 8 cm, respectively. The ambient temperature during the experiment was 10 °C. The working condition numbers for 36 working conditions were EGK1-EGK36, as shown in Table 2. It can be calculated that all operating conditions are well-ventilated compartment fires.

**Table 2.** Experimental conditions of narrow and restricted space window plume model.

| Working Condition | The Fire HRR $Q$ (Kw) | Opening Size | | The Distance $D$ (cm) |
|---|---|---|---|---|
| | | Height $H$ (m) | Width $W$ (m) | |
| EGK1 | | | 0.5 | 32 |
| EGK2 | | 0.4 | 0.3 | 32 |
| EGK3 | | | 0.2 | 32 |
| EGK4 | | | 0.5 | 24 |
| EGK5 | | 0.4 | 0.3 | 24 |
| EGK6 | | | 0.2 | 24 |
| EGK7 | 6.81 | | 0.5 | 16 |
| EGK8 | | 0.4 | 0.3 | 16 |
| EGK9 | | | 0.2 | 16 |
| EGK10 | | | 0.5 | 8 |
| EGK11 | | 0.4 | 0.3 | 8 |
| EGK12 | | | 0.2 | 8 |
| EGK13 | | | 0.5 | 32 |
| EGK14 | | 0.4 | 0.3 | 32 |
| EGK15 | | | 0.2 | 32 |
| EGK16 | | | 0.5 | 24 |
| EGK17 | | 0.4 | 0.3 | 24 |
| EGK18 | 9.08 | | 0.2 | 24 |
| EGK19 | | | 0.5 | 16 |
| EGK20 | | 0.4 | 0.3 | 16 |
| EGK21 | | | 0.2 | 16 |
| EGK22 | | | 0.5 | 8 |
| EGK23 | | 0.4 | 0.3 | 8 |
| EGK24 | | | 0.2 | 8 |
| EGK25 | | | 0.5 | 32 |
| EGK26 | | 0.4 | 0.3 | 32 |
| EGK27 | | | 0.2 | 32 |
| EGK28 | | | 0.5 | 24 |
| EGK29 | | 0.4 | 0.3 | 24 |
| EGK30 | | | 0.2 | 24 |
| EGK31 | 17.84 | | 0.5 | 16 |
| EGK32 | | 0.4 | 0.3 | 16 |
| EGK33 | | | 0.2 | 16 |
| EGK34 | | | 0.5 | 8 |
| EGK35 | | 0.4 | 0.3 | 8 |
| EGK36 | | | 0.2 | 8 |

*3.2. Experimental Results*

Table 3 shows the characteristic length $L_2$ calculated by Equation (6) when there is no vertical retaining wall outside the building opening. The vertical temperature was measured using thermocouples inside the confined space. Figure 5 shows the relationship between the vertical plume temperature inside the confined space and the distance $D$ ($Z$ represents the height to the ground). Due to the same conclusions obtained, this study only displays the temperature distribution of conditions EGK1-EGK12. When the distance $D$ is greater than $L_2$, the temperature inside the confined space remains basically unchanged as $D$ decreases. When the distance $D$ is less than $L_2$, the temperature inside the confined

space increases sharply as *D* decreases. The main reason is that when *D* is less than $L_2$, the vertical retaining wall restricts the plume's entrainment of air radially (radial to the direction of the building facade), affecting the decrease in temperature.

**Table 3.** Value of characteristic length $L_2$ with no vertical retaining wall outside the building.

| The fire HRR $Q$ (kW) | 6.81 | | | 9.08 | | | 17.84 | | |
|---|---|---|---|---|---|---|---|---|---|
| Opening width $W$ (m) | 0.5 | 0.3 | 0.2 | 0.5 | 0.3 | 0.2 | 0.5 | 0.3 | 0.2 |
| $L_2$ (m) | 0.199 | 0.154 | 0.148 | 0.190 | 0.159 | 0.136 | 0.163 | 0.129 | 0.109 |

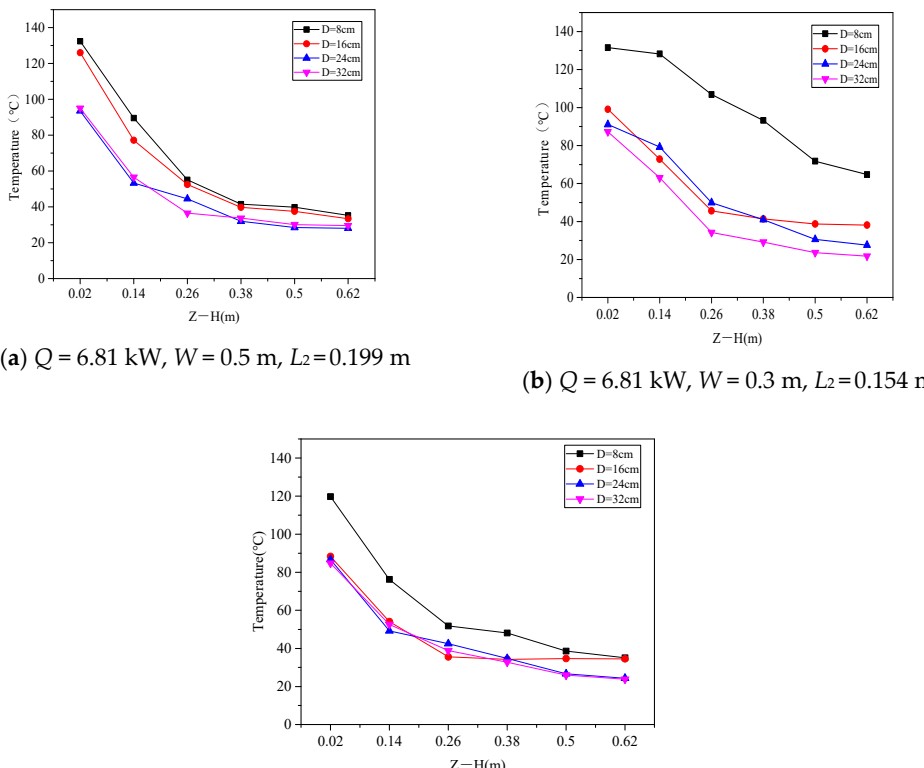

(**a**) $Q$ = 6.81 kW, $W$ = 0.5 m, $L_2$ = 0.199 m

(**b**) $Q$ = 6.81 kW, $W$ = 0.3 m, $L_2$ = 0.154 m

(**c**) $Q$ = 6.81 kW, $W$ = 0.2 m, $L_2$ = 0.148 m

**Figure 5.** Relationship between vertical plume temperature and height.

When the distance *D* is less than $L_2$, the experimental data of different fire HRR were summarized and fitted to obtain the relationship between temperature rise and vertical height inside the confined space. $Q_1$ of the virtual ignition source varies with different opening sizes, and the average value is taken in this study, as shown in Figure 6. It can be observed that $\Delta T$ and $z$ are inversely proportional, which is consistent with Equation (41).

To eliminate the influence of fire HRR and obtain the general law of temperature rise and height, the vertical plume temperature rise and fire HRR are dimensionless, as shown in Figure 7. The fitting results are as follows:

$$\Theta/(Q_1^*)^{2/3} = \frac{\frac{\Delta T_\infty}{T_\infty}}{\left(\frac{Q_1}{T_\infty \rho_\infty c_P \sqrt{g}}\right)^{2/3}} = 0.63(\alpha D)^{-2/3} z^{-1} \tag{42}$$

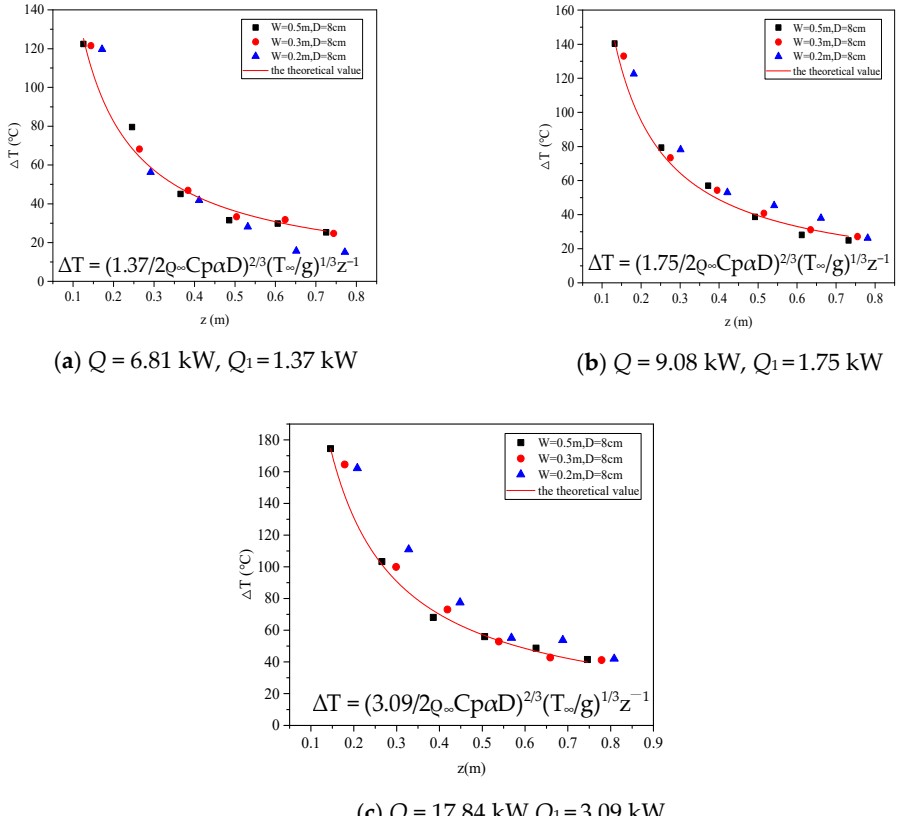

(**a**) $Q = 6.81$ kW, $Q_1 = 1.37$ kW

(**b**) $Q = 9.08$ kW, $Q_1 = 1.75$ kW

(**c**) $Q = 17.84$ kW, $Q_1 = 3.09$ kW

**Figure 6.** Relationship between temperature rise and height in a confined space.

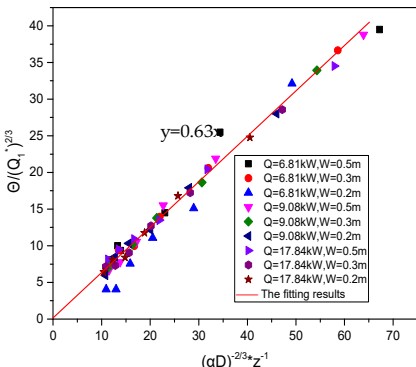

**Figure 7.** Non-dimensional value $\Theta/(Q_1{}^*)^{2/3}$ versus $(\alpha D)^{-2/3}z^{-1}$.

## 4. Numerical Simulation

This chapter mainly introduces the numerical simulation research on the influence of the size of enclosed spaces on the temperature distribution characteristics of window plumes in well-ventilated compartment fires. Among them, Section 4.1 introduces the construction of the numerical model, and Section 4.2 introduces the independent analysis of the grid. Section 4.3 verifies the reliability of numerical simulation.

### 4.1. Fire Scenarios

Fire Dynamics Simulator (FDS) is usually used to simulate fire environment [31], a specialized software for fire dynamics simulation that was developed based on the theory of large eddy simulation (LES).

After the initial stage of the fire, the range of spread is approximately 4 floors high. Based on the spread of the fire and the limitations of computer resources, the length, width, and height of typical high-rise buildings in this study are 6 m, 20 m, and 32 m, respectively,

as shown in Figure 8. The length, width, and height of the room on fire are 6 m, 20 m, and 5 m, respectively, and the window height is 5 m. The width of the window can be adjusted according to different research conditions. A total of three window widths $W$ were designed, which were 10 m, 4 m, and 2 m, in descending order. A vertical retaining wall was installed on the outer side of the building facade. The distance $D$ between the building facade and the vertical retaining wall can also be adjusted. In this study, a total of 6 distances, $D$, were set, which were 10 m, 8 m, 6 m, 3 m, 2 m, and 1 m. The fire source was located in the middle of the burning room, with an area of 1 m × 1 m. According to relevant regulations, the fire HRR of common buildings were calculated, and in order to enrich data, the fire HRR $Q$ were 10 MW and 6 MW respectively. Working conditions are as shown in Table 4. It can be calculated that all conditions were well-ventilated compartment fires. In the simulation, the ambient temperature was set to 20 °C, without considering the influence of surrounding wind. To reduce the influence of boundaries on the simulation results, an an extension of 1.5 m was made in the length direction according to the different distances between retaining wall and the building facade in the simulation. For example, when D is 10 m, the length, width, and height of the entire calculation domain are 18 m, 20 m, and 32 m, respectively. The boundaries of the grid are all set to open. The soot yield applied for the combustion model is 0.024.

**Table 4.** Simulation table.

| Working Condition | Fire HRR $Q$ (MW) | Window Height $W$ (m) | The Distance $D$ (m) | Working Condition | Fire HRR $Q$ (MW) | Window Height $W$ (m) | The Distance $D$ (m) |
|---|---|---|---|---|---|---|---|
| SGK1 | | | 1 | SGK19 | | | 1 |
| SGK2 | | | 2 | SGK20 | | | 2 |
| SGK3 | | | 3 | SGK21 | | | 3 |
| SGK4 | | 10 | 6 | SGK22 | | 10 | 6 |
| SGK5 | | | 8 | SGK23 | | | 8 |
| SGK6 | | | 10 | SGK24 | | | 10 |
| SGK7 | | | 1 | SGK25 | | | 1 |
| SGK8 | | | 2 | SGK26 | | | 2 |
| SGK9 | 10 | 4 | 3 | SGK27 | 6 | 4 | 3 |
| SGK10 | | | 6 | SGK28 | | | 6 |
| SGK11 | | | 8 | SGK29 | | | 8 |
| SGK12 | | | 10 | SGK30 | | | 10 |
| SGK13 | | | 1 | SGK31 | | | 1 |
| SGK14 | | | 2 | SGK32 | | | 2 |
| SGK15 | | 2 | 3 | SGK33 | | 2 | 3 |
| SGK16 | | | 6 | SGK34 | | | 6 |
| SGK17 | | | 8 | SGK35 | | | 8 |
| SGK18 | | | 10 | SGK36 | | | 10 |

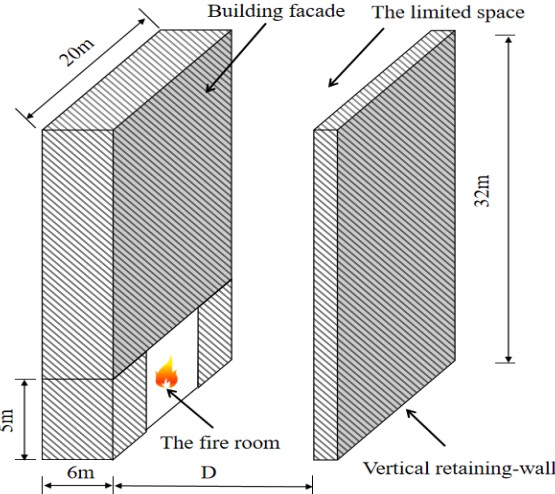

**Figure 8.** Narrow and confined space model.

In order to better obtain the temperature and velocity distribution inside the confined space, temperature, and velocity measurement points have been set up in the confined space. A total of twelve measurement points were set up inside the confined space perpendicular to the direction of the building facade, with a spacing of 0.5 m between the first four columns and 1 m between the last 8 columns. Each column has a total of thiry-two measurement points in the vertical direction, with a spacing of 1 m. As the distance *D* decreases, the number of measurement points in the direction perpendicular to the building facade also decreases. As shown in Figure 9.

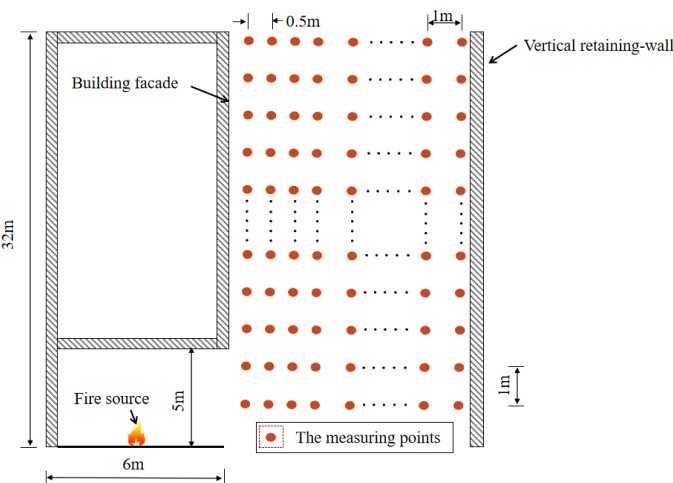

**Figure 9.** Arrangement of internal measuring points.

*4.2. Grid Analysis*

Usually, the method of calculating the characteristic diameter *D\** of the fire source is used to determine the mesh accuracy in order to preliminarily and judge the flow field. In the user manual, the characteristic diameter *D\** of the combustion source is calculated based on the grid size $\delta_x$. The ratio $(D^*/\delta_x)$ is used as a criterion for selecting the size of the grid. When the range of this ratio is between 4 and 16, the simulation calculation results are relatively good, and the range of grid size is from 0.0625 *D\** to 0.25 *D\** [32]. The corresponding relationship is as follows:

$$D^* = \left(\frac{Q}{\rho_\infty C_p T_\infty \sqrt{g}}\right)^{\frac{2}{5}} \tag{43}$$

IIn current study, a grid size of 0.2 m (approximately 0.01 *D\** and 0.085 *D\** for 6 MW and 10 MW, respectively) was considered reasonable. To justify the rationality of grid size, we selected other grid sizes of 0.1 m × 0.1 m × 0.1 m, 0.3 m × 0.3 m × 0.3 m, 0.4 m × 0.4 m × 0.4 m for comparison. Taking an typical 6 MW and 10 MW fire in the building mentioned above (the length, width, and height are 6 m, 20 m, and 32 m, respectively) as an example. The length, width, and height of the room on fire are 6 m, 20 m, and 5 m respectively. The distance D between the building facade and the vertical retaining wall is 10 m. The numerical simulation domain's length, width, and height are 18 m, 20 m, and 32 m, respectively. The vertical temperature at a distance of 0.8 m from the building facade is shown in Figure 10. We observe that the numerical result is independent of the mesh size at less than 0.2 m, there is no significant improvement in simulation results for 0.1 m. Therefore, the grid size of 0.2 m is considered appropriate.

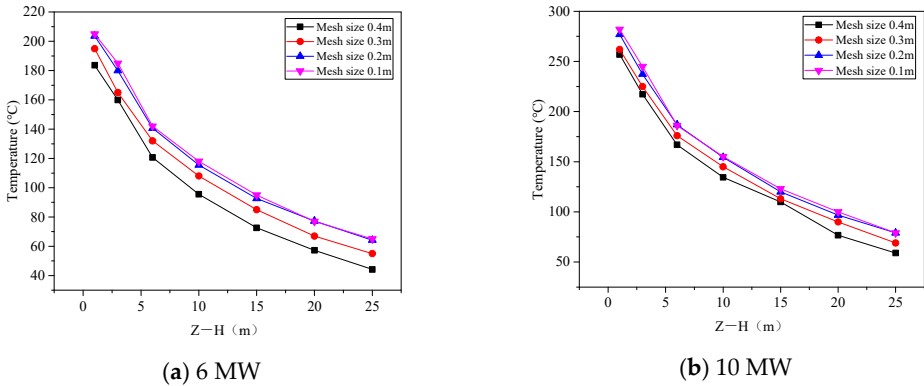

(**a**) 6 MW  (**b**) 10 MW

**Figure 10.** Vertical temperature at 0.8 m away from the building facade.

### 4.3. Validation of Modelling

In order to verify the reliability of FDS for this study, this article uses numerical simulation software FDS to establish a 1:1 numerical model with the experimental platform, and other parameters are basically consistent with those in the experiment. The location and characteristics of the fire source, as well as the setting of environmental parameters, are consistent with the relevant parameters of the experiment. The numerical simulation model and the setting of some parameters in the model are shown in Figures 11 and 12, respectively. Due to the consistent conclusions, this presentation takes working conditions 1 and 7 as an example. Temperature is the most critical parameter for studying plume propagation. As shown in Figure 13, through comparison, it was found that the simulation results of the same size of the vertical temperature inside the confined space are basically consistent with the experimental results (EGK represents experimental data and ESGK represents numerical simulation data), indicating that numerical simulation can effectively combine with the experiment and make up for the shortcomings of the experiment.

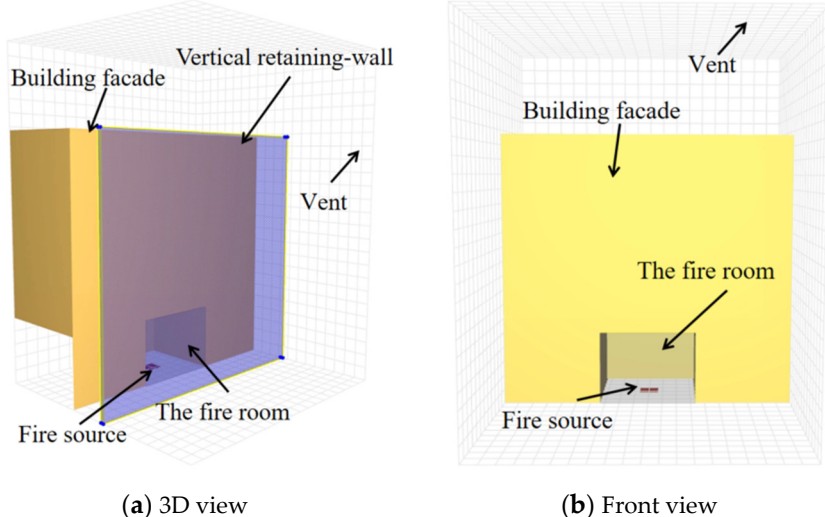

(**a**) 3D view  (**b**) Front view

**Figure 11.** 1:1 numerical simulation model with experimental platform.

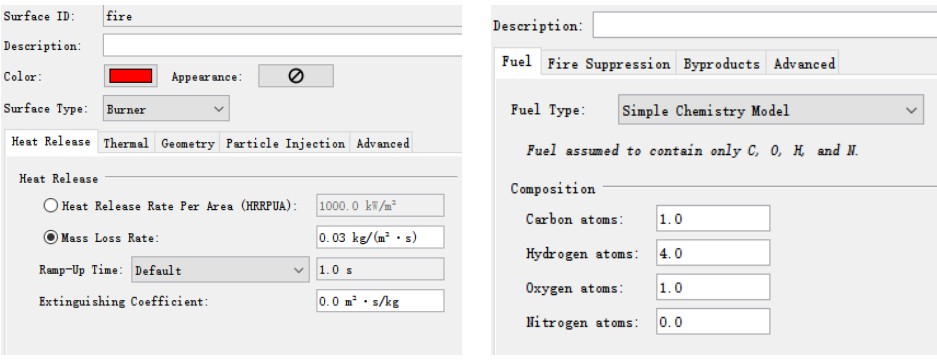

(**a**) Mass flow rate of fuel          (**b**) The composition of fuel

**Figure 12.** Setting of the fire source in the FDS numerical model.

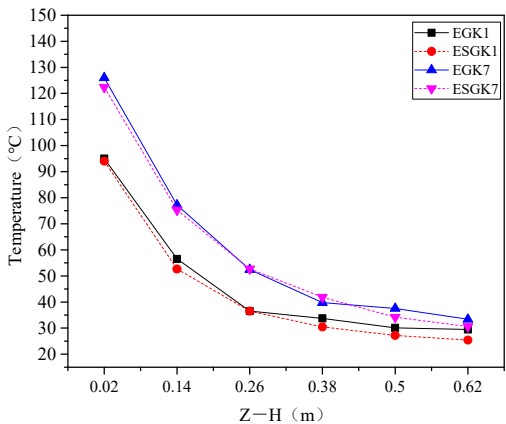

**Figure 13.** Comparison of numerical simulation and experimental temperature Data.

## 5. Results

This chapter mainly introduces the main results obtained from numerical simulation. Among them, Section 5.1 introduces the radial (perpendicular to the building facade) temperature distribution; Section 5.2 introduces the vertical temperature distribution.

### 5.1. Radial (Perpendicular to the Building Facade) Temperature Distribution

Figures 14–16 show the radial temperature distribution and window plume propagation at different heights of the central section of the internal plume in a confined space with a distance $D$ ($D = 10$ m) (due to the same conclusions obtained, this study only shows the working condition of $Q = 10$ MW). It can be observed that when the distance $D$ is large, the radial temperature inside the confined space can approximate a Gaussian distribution, and the spread of the plume varies under different windows. When the window width is large, the maximum vertical temperature of the plume is close to the wall surface, which is known as the attached plume. When the window width is small, due to the large horizontal momentum when smoke overflows from the window, the maximum vertical temperature of the plume is far from the wall, which is a non-attached plume, is consistent with previous research conclusions [33–36].

According to Equation (6), the numerical summary of the characteristic length $L_2$ under free working conditions without vertical retaining walls outside the building window is shown in Table 5.

**Table 5.** Value of characteristic length $L_2$ under free working conditions.

| **Fire HRR (MW)** | **10** | | | **6** | | |
|---|---|---|---|---|---|---|
| Window Width $W$ (m) | 10 | 4 | 2 | 10 | 4 | 2 |
| $L_2$ (m) | 2.18 | 1.53 | 1.12 | 1.97 | 1.45 | 1.06 |

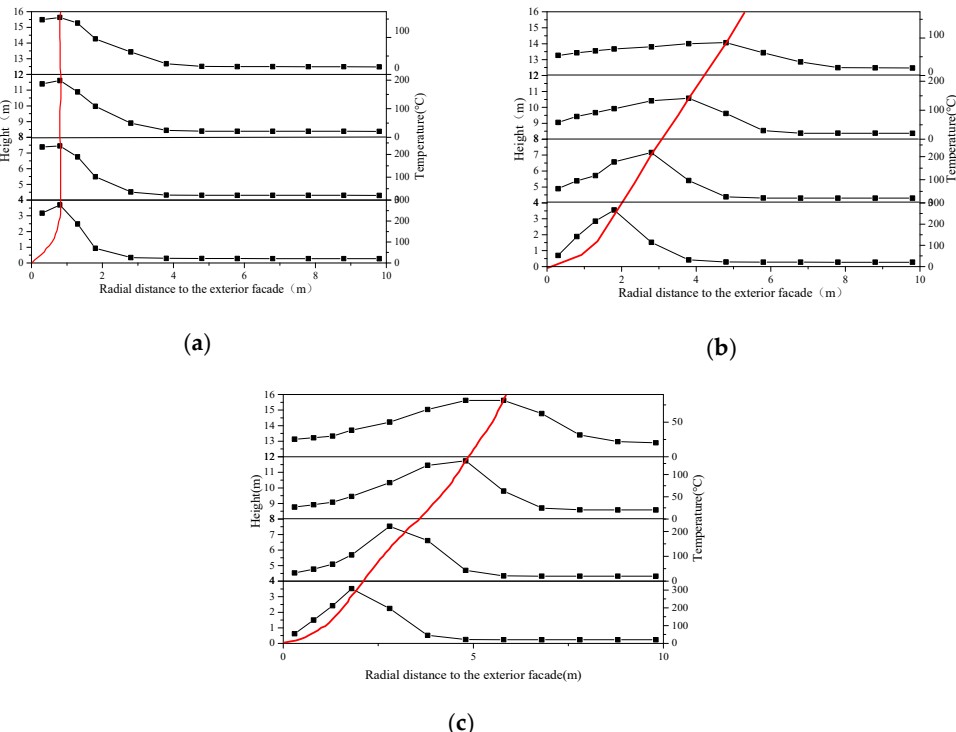

**Figure 14.** Plume temperature distribution and smoke spread at different heights when the distance *D* is 10 m. (**a**) *W* = 10 m; (**b**) *W* = 4 m; (**c**) *W* = 2 m.

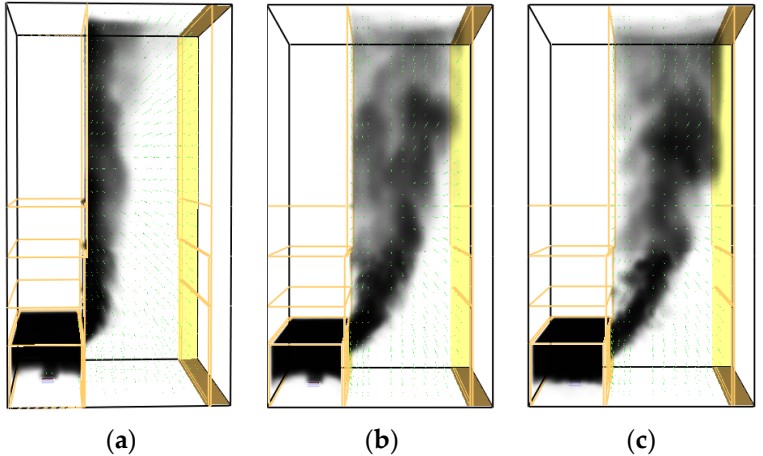

**Figure 15.** Smoke spread of different window widths when the distance *D* is 10 m. (**a**) *W* = 10 m; (**b**) *W* = 4 m; (**c**) *W* = 2 m.

Figure 17 shows the comparison of temperature distribution in the radial direction of the plume under different distances of the retaining walls and building facades. Due to the same conclusions obtained, this study only displays the temperature distribution of SGK1-SGK 6. It can be observed that when the distance *D* is greater than $L_2$, as *D* decreases, the radial temperature of the plume does not change much and still follows the Gaussian distribution model. When *D* is less than $L_2$, both the outer side of the ejecting plume and the side near the building facade are affected by air entrainment, so the radial temperature distribution inside the confined space will tend to be consistent, and the temperature of the plume will only change vertically.

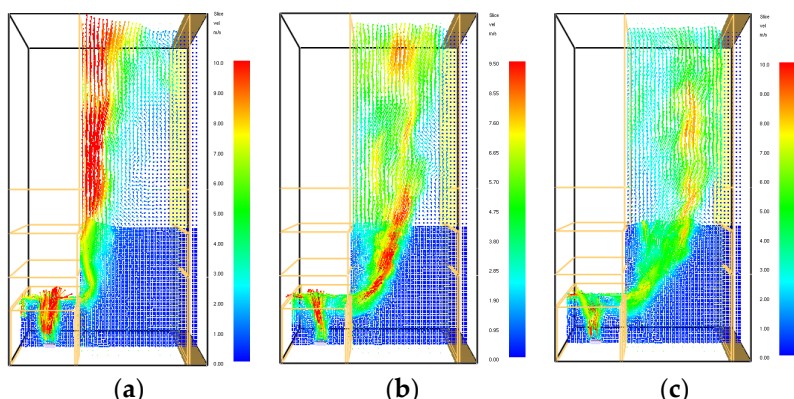

**Figure 16.** Velocity distribution of plumes on different window widths when the distance *D* is 10 m. (**a**) *W* = 10 m; (**b**) *W* = 4 m; (**c**) *W* = 2 m.

(**a**) *D* = 10 m

(**b**) *D* = 8 m

(**c**) *D* = 6 m

(**d**) *D* = 3 m

(**e**) *D* = 3 m

(**f**) *D* = 1 m

**Figure 17.** Plume temperature distribution under the different distances of the retaining walls and building facades.

Figure 18 shows the spread of smoke under different spaces of the retaining wall and building facades. It can be observed that when $D$ is greater than $L_2$, as the spaces decrease, it has almost no effect on the air around the plume entrainment. When $D$ is less than $L_2$, the retaining wall is close to the outer boundary of the plume, which affects the air entrainment of the plume. This also verifies the temperature distribution law obtained in the previous text and verifies the accuracy of using Equation (6) to calculate $L_2$.

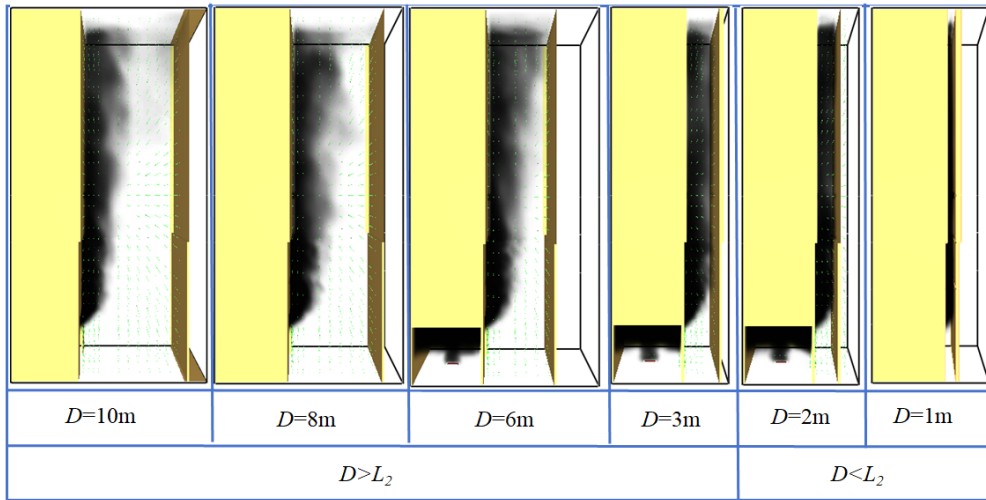

| $D$=10m | $D$=8m | $D$=6m | $D$=3m | $D$=2m | $D$=1m |
|---|---|---|---|---|---|
| $D>L_2$ | | | | $D<L_2$ | |

**Figure 18.** Smoke spread distribution under the different distances of the retaining wall and building facades.

### 5.2. Vertical Temperature Distribution

The vertical temperature distributions in Figures 19 and 20 show the effect of distance $D$ on temperature distribution at various heights in a confined space at a certain radial distance. In order to more clearly see the impact of distance $D$ on temperature distribution in the confined space, this study takes the temperature distribution at heights of 0.3 m and 0.8 m (radial distance) from the outer facade of the central section of the plume for analysis. Due to the same conclusion, this study only takes SGK1-SGK18 as an example for analysis. Through comparison, it can be found that when the distance $D$ is greater than $L_2$, the temperature change trend of each point in the confined space increases slowly with the decrease of $D$. When $D$ is less than $L_2$, the temperature of each point in the confined space increases sharply with the decrease of $D$. The reason for the analysis is that when $D$ is large, the impact on the horizontal entrainment of surrounding air by the plume is relatively small. When $D$ is small, the plume seriously affects the horizontal entrainment of surrounding air, thereby affecting the reduction of smoke temperature.

As mentioned above, when $D$ is less than $L_2$, the radial entrainment will be almost completely limited, considering only the entrainment on both sides. The plume inside the confined space is considered to be generated by a virtual ignition source. According to the derivation, the temperature rise at different heights in the confined space varies with height, as shown in Equation (41).

Based on the derived plume model of the internal ignition source in a confined space, simulation data of different ignition source powers were summarized and fitted when the distance $D$ was less than $L_2$. The relationship between the temperature rise $\Delta T$ inside the confined space and the height $z$ to the virtual ignition source was obtained, as shown in Figure 21. It also can be observed that $\Delta T$ and $z$ are inversely proportional, which is consistent with Equation (41).

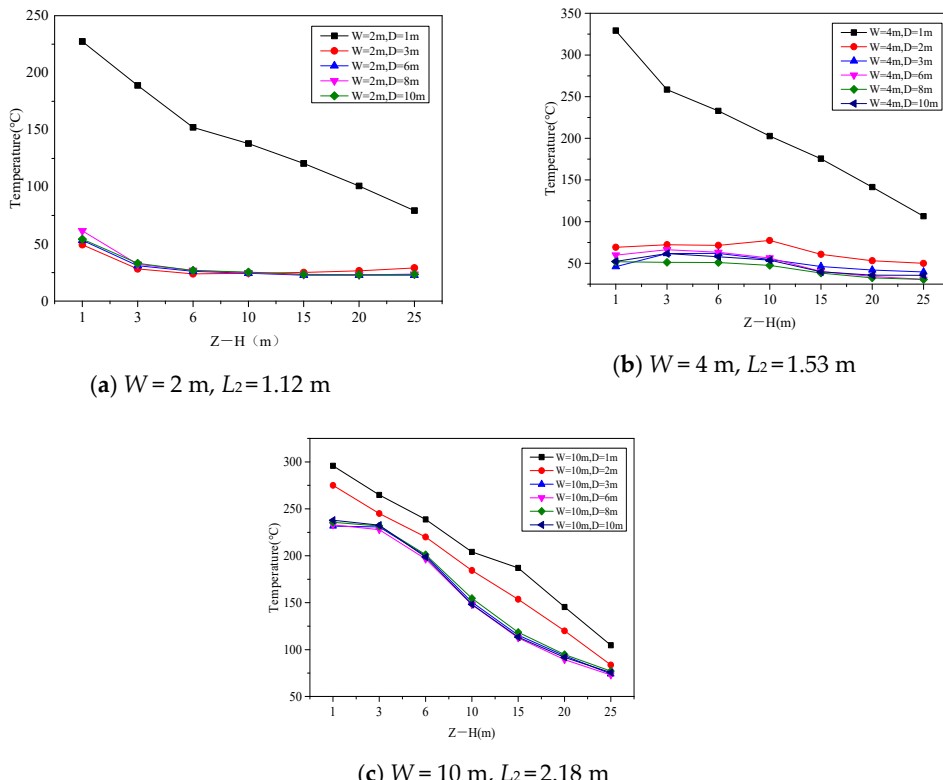

(**a**) *W* = 2 m, *L₂* = 1.12 m

(**b**) *W* = 4 m, *L₂* = 1.53 m

(**c**) *W* = 10 m, *L₂* = 2.18 m

**Figure 19.** Temperature distribution at different heights of a radial distance of building facade under different confined spaces (radial distance 0.3 m).

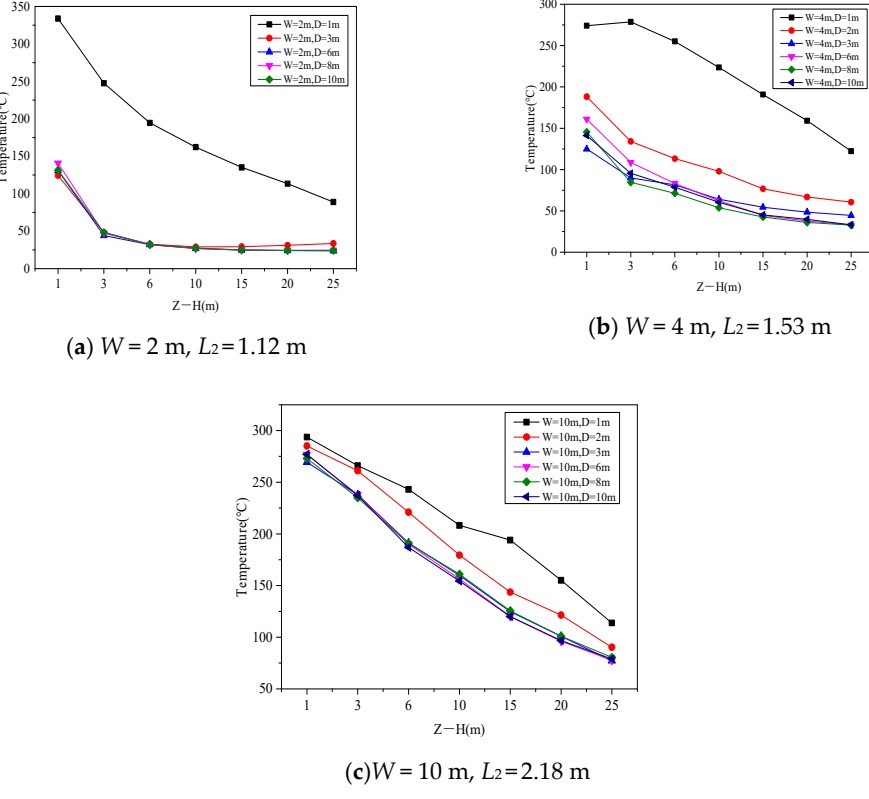

(**a**) *W* = 2 m, *L₂* = 1.12 m

(**b**) *W* = 4 m, *L₂* = 1.53 m

(**c**)*W* = 10 m, *L₂* = 2.18 m

**Figure 20.** Temperature distribution at different heights of a radial distance of building facade under different confined spaces (radial distance 0.8 m).

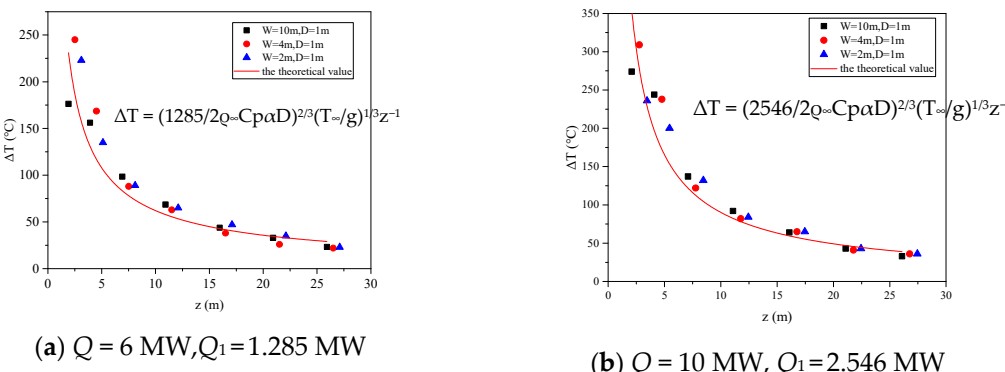

**(a)** $Q$ = 6 MW, $Q_1$ = 1.285 MW

**(b)** $Q$ = 10 MW, $Q_1$ = 2.546 MW

**Figure 21.** Temperature rise at different heights in confined space.

Summarizing all data and the dimensionless model, the vertical plume temperature rise and fire HRR, as shown in Figure 22. The fitting results are as follows:

$$\Theta/(Q^*)^{2/3} = \frac{\Delta T/T_\infty}{(Q^*)^{2/3}} = 1.2(\alpha D)^{-2/3}z^{-1} \tag{44}$$

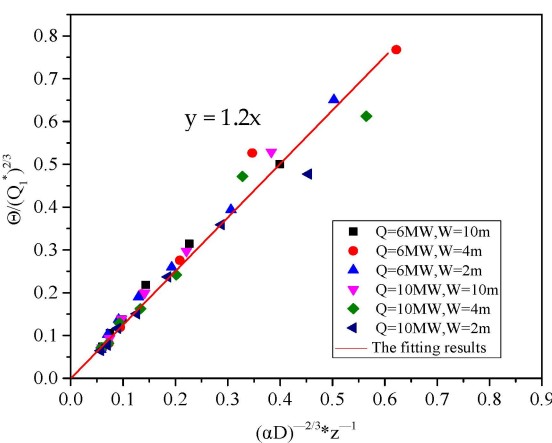

**Figure 22.** Non-dimensionless value $\Theta/(Q_1^*)^{2/3}$ versus $(\alpha D)^{-2/3}z^{-1}$.

It can be found that the trend of the relationship between the dimensionless temperature rise obtained from numerical simulation and the dimensionless fire HRR changing with height is basically consistent with the trend obtained from the experiment. The difference between the coefficients of Equations (42) and (44) may be due to the significant influence of the surrounding environment during the experiment. The default parameters for numerical simulation are unchanged for the environment. There are still some differences from the actual situation.

## 6. Conclusions and Discussion

This paper conducts experimental and numerical simulation research on the characteristics of window plume propagation in confined spaces under fuel-controlled combustion and, based on this, explores the degree of harm of fire propagation. The main conclusions can be drawn as follows:

1.  There is also a critical distance $L_2$ in confined spaces under well-ventilated compartment fires, which can provide a basis for the width of confined spaces. When the distance $D$ is greater than $L_2$, the radial temperature inside the confined space can approximate a Gaussian distribution. As the distance $D$ decreases, the radial temperature change of the plume is not significant, and the temperature increase at each point is relatively slow. When $D$ is less than $L_2$, the radial temperature distribution inside

the confined space will tend to be consistent, and the temperature at each point in the confined space will sharply increase as *D* decreases.

2.  When the distance *D* is less than $L_2$, based on the internal ignition source plume model of the confined space, the relationship between dimensionless temperature rise and vertical height can be obtained. The acquisition of this relationship can provide guidance for determining the vertical spread of similar building fire hazards.

3.  The above conclusion can provide a basis for the fire prevention design of glass curtain wall buildings and narrow street valleys. However, this paper did not consider the influence of the location of the fire source in the room on the temperature distribution characteristics of the internal window plume in the confined space. Further research can be conducted on the impact of the fire source location on the subsequent window plume in larger office buildings.

**Author Contributions:** Conceptualization, Q.D. and Y.L.; methodology, J.L.; software, D.X.; valida­tion, Y.L., F.X. and Z.S.; data curation, Q.D.; writing—original draft preparation, Q.D.; writing—review and editing, Q.D.; project administration, Q.D., Y.L. and J.L.; funding acquisition, J.L. All authors have read and agreed to the published version of the manuscript.

**Funding:** This research was funded by the Beijing Natural Science Foundation, grant number L140002 and 8172006.

**Institutional Review Board Statement:** Not applicable.

**Informed Consent Statement:** Not applicable.

**Data Availability Statement:** The datasets used during the current study are available from the corresponding author on reasonable request.

**Conflicts of Interest:** The authors declare no conflicts of interest.

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
