# Peer review of "The Influence of Confined Space Size on the Temperature Distribution Characteristics of Internal Window Plume from Well-Ventilated Compartment Fires"

_fire, doi:10.3390/fire7050158_

Round 1

Reviewer 1 Report

Comments and Suggestions for Authors

Apart from minor comments such as:

on line 420 there is: Eqs. 43 and 44, should be: Eqs.42 and 44

on line 421 there is: comma, it should be: dot,

I have two main comments:

1. In point 4.2. Grid analysis, description is unclear.

2. In point 4.3. Verification of modeling, description is insufficient. Changing the geometric scale of the model brings with it changes in other similarity scales. The authors did not clearly describe whether the numerical simulations are performed on the same scale as the experimental model? If so, please write it clearly. Otherwise, there is a problem of criterion numbers describing the change in the scales of other parameters describing the phenomenon.

To correct or supplement the above points, I suggest the authors of the thesis to refer to publications that address exactly the same problem as presented in the article below. One of the articles presents the results of measurements and simulations on exactly the same geometric scale of 1:8:

https://www.sciencedirect.com/science/article/abs/pii/S1359431117322810

https://journals.plos.org/plosone/article?id=10.1371/journal.pone.0225120

https://link.springer.com/chapter/10.1007/978-981-32-9139-3_9

Author Response

Dear expert, first of all, thank you for taking the time out of your busy schedule to provide valuable suggestions for the article. Your comments are very helpful for improving this article. Based on your comments, I have also made revisions and responses point-by-point. Now I am submitting the responses and revised file to you for your review.Please see the attachment.

Reviewer 2 Report

Comments and Suggestions for Authors

In this article, the authors performed a numerical study on the effects of confined space on the temperature and smoke of a window plume in well-ventilated compartments fires. The model was validated against small scale experiments. Compartment fire dynamics is an important topic as it is the foundation of fire safety engineering. Overall the article is well written but the novelty of this study can be further highlighted as the FDS fire model is widely applied for both research and industry. Also the results and key findings can be strengthen by more in-depth discussions.  Therefore, it is recommended that this article undergo major revision before considered for publication.

Specific comments:
1. as highlighted above, the novelty of this study can be further highlighted in the introduction. There are numerous existing experimental and numerical studies on comparment fires and windows plume with similar scenarios to this manuscript (e.g. double skin facade fires). 

2. Also, i am not very convinced of the merits of studying well-ventilated conditions because the majority of existing literature focuses on underventilated scenarios. One of the reasons for the focus towards underventilated conditions is because it is much more challenging to model in CFD and involves more complex behaviours. It is recommended the authors to strengthen this discussion with more references.

3. This article needs a nomenclature section or the author needs to make sure all the symbols are well defined. it will improve the overall readibility of the article.

4. in Fig 3.  the fire location and the distance D should also be indicated in (a).

5. In 3.2 experimental results, the results from Fig 6 and 7 can be discussed in more detail. For example, why in Fig 5(b), the D=8cm case show such different temperature profile compared to the other cases? Also can discuss the significance of the non-dimensional analysis in Fig 6-7.

6. In 4.1, the authors should provide more details on computational domain and the boundary conditions. Was the vertical retaining wall also included inside the domain? What is the soot yield applied for the combustion model? This value will have noticeable effect on the temperature and smoke predictions

7. Section 4.3 is more validation than verification. Please revise the heading accordingly. Is there a particular reason why EGK1 and EGK7 were chosen for the validation? What is the overall discrepancy between the experiments and simulation?

8. In section 5, the figures can be improved, the aspect ratio of the plots, legend and labels seems to be skewed and inconsistent. 

9. The authors can consider also presenting more CFD specific results such as velocity and temperature contours to provide more visual to understand the fire plume characterisitics. Currently, the majority of the results are focused on the temperature measurement points only.

9. The discussion on the smoke spread distribution is lacking. Perhaps the authors can consider either analysing the soot mass fraction or smoke visibility as measurement points or 2D contours.

10. In the conclusion, the key findings mainly focuses on the ratio between D and L2. L2 is referred to as the critical distance, and in the main article, it is also refered to as the characterisitic length. Perhaps the authors can provide more discussion on the physical significance of this parameter and how it relates to the fire plume.

Comments on the Quality of English Language

Overall, the article is well written with only minor instances of awkward grammar that could be further improved through thorough proofreading.

Author Response

(The authors gave the same response as above.)

Reviewer 3 Report

Comments and Suggestions for Authors

In accordance with the proofreading criteria of the publisher, I prepared a reviewer’s report, which would be as follows:

I. General opinion.

In my opinion the content of the proposed paper on high level meets the objectives of the journal.

Using the scientific methods (numerical simulation and experimental method) for the preparation of the case study applied in accordance with the author’s scientific objectives resulted useful scientific achievements.

The references used in literature review and the main chapters are relevant and assist the reader to understand the authors proposals. The illustrations used are regular and correct.

II. Detailed report

Abstract: The abstract meets the requirements of content for the research papers, however the authors should indicate the main "conclusions or interpretations" of their article.

1. Introduction. At the end of Introduction (Section 1), the main conclusions should be highlighted as well in more detail. The literature review is clear and adequately represents the scientific situation of the chosen theme and the novelties added by the authors.

The "Materials and Methods" used in the present article can be found in Sections 2-4, which are well structured and adequately transparent.

The figures and tables created by the authors well support the information the authors want to convey to us. However, it should be considered, that at the beginning of these Sections as an introduction, the main content of the subsections should be briefly described.

5. Results and Discussion. The results of the experiments and the simulation were presented correctly, and are well supported from a methodological point of view. Anyway, this chapter does not contain a „Discussion” and it would be advisable to summarize the research results in a separate subsection at the end of the Section.

6. Conclusions. It is recommended to specify shortly the possibilities of international theoretical and practical applicability of the presenty research, as well as the possible directions of future research projects. 

 Based on the above, after a revision, I suggest the publication of reviewed article.

Author Response

(The authors gave the same response as above.)

Round 2

Reviewer 2 Report

Comments and Suggestions for Authors

The authors have adequately addressed most of the issues raised in the comments. Regarding the issues on smoke spread, considering the main focus of this manuscript is on temperature profiles, the revisions made by the authors have brought the paper to an acceptable form.

Author Response

Dear expert, first of all, thank you again for taking the time out of your busy schedule to provide valuable suggestions for the article, and also thank you for your recognition of this article. Your comments are very helpful for improving this article and other future researches. Wishing you all the best in your work and good health.